# Effect of Gluten Composition in Low-Allergy O-Free Wheat Flour on Cookie-Making Performance Compared with Flours with Different Gluten Strengths

**DOI:** 10.3390/foods12203843

**Published:** 2023-10-20

**Authors:** Nayeon Baek, Yujin Moon, Jeongeon Kim, Meera Kweon

**Affiliations:** 1Department of Nutrition Education, Pusan National University, Busan 46241, Republic of Korea; qorsk100@pusan.ac.kr; 2Department of Food Science and Nutrition, Pusan National University, Busan 46241, Republic of Korea; amoebacul@pusan.ac.kr (Y.M.); wjddjs8@pusan.ac.kr (J.K.); 3Kimchi Research Institute, Pusan National University, Busan 46241, Republic of Korea

**Keywords:** low-allergy wheat, SDS-PAGE, thermal and pasting property, mixogram, cookie

## Abstract

The increasing demand for allergen-free and reduced-allergen foods has led to an investigation into the potential use of O-free wheat, a low-allergy wheat cultivar, in cookie production. This study focused on assessing the gluten composition of O-free flour and comparing its suitability for cookie making in comparison to flours with varying gluten strengths. Several analyses were conducted, including gluten composition, solvent retention capacity (SRC), thermal and pasting properties, dough-mixing characteristics, and cookie-making performance. The gluten composition of O-free flour by SDS-PAGE confirmed the absence of ω-gliadins and the reduced levels of low-molecular-weight glutenins and γ-gliadins. The SRC values of O-free flour fell between the flours with weak and medium gluten strengths. While thermal and pasting properties showed significant differences in sucrose solution but not across flour types, indicating similar starch structures, mixograms displayed distinct variations influenced by both sucrose solution and flour type, highlighting the importance of gluten quality and composition. Cookies made with O-free flour demonstrated similarities to those produced with weak gluten flour, known for their favorable cookie characteristics. This study emphasizes the significant influence of flour gluten composition on cookie-making performance and advocates for the adoption of O-free flour in the development of allergy-friendly cookies.

## 1. Introduction

Food allergies are hypersensitivity reactions caused by an abnormal immune response to specific foods [1]. The prevalence of food allergies is increasing worldwide, including domestically [2]. Cereals containing gluten, such as wheat, are frequently associated with allergic reactions in Korean adults. Wheat allergies pose a particular challenge as they can lead to wheat-dependent exercise-induced anaphylaxis (WDEIA), which can be fatal [3]. To address this, the Rural Development Administration in Korea has recently developed a specialty wheat variety called O-free, which does not contain the main allergens such as ω-5-gliadins, making wheat consumption safe for individuals with wheat allergies [4]. Previous research has investigated the performance of O-free flour in bread and noodle production, broadening its potential applications [5,6,7]. However, there is a lack of in-depth studies on the performance of O-free flour in cookie making. 

In cookie production, sucrose, a primary ingredient, suppresses gluten development during mixing and delays starch gelatinization and pasting during baking, resulting in cookies with large diameters and small stack heights [8,9,10,11]. Numerous studies have examined the effects of wheat flour protein content and composition on cookie quality [12,13,14,15,16,17]. Protein content negatively affects cookie quality, reducing diameter and increasing hardness [14,16]. Gluten, a component of wheat protein, negatively affects cookie diameter by limiting the dough spread time [13] and preventing collapse through gluten entanglement and/or cross-linking during baking [12]. Moreover, the deletion of *Glu-D1*-encoded high-molecular-weight glutenin subunits resulted in poor gluten quality but improved cookie quality [15]; subunits 2 + 12 among the high-molecular-weight glutenin subunits encoded by the *Glu-D1* locus positively influenced cookie quality [16]. 

Sulfur-containing amino acid residues in proteins play a crucial role in the formation of disulfide bonds through free thiols during gluten development. Gliadins and glutenins, excluding ω-gliadin, contain these sulfur amino acid residues, which impact gluten development and strength, ultimately leading to reduced cookie spread [18,19]. However, previous study by Uthayakumaran et al. [20] has reported the effects of gliadin alone on cookie quality, such as increased diameter and thickness, softer texture, and enhanced dough extensibility of the γ-gliadin fraction, which was expected to improve cookie quality. Barak et al. [21] also demonstrated a significant positive correlation between the cookie spread factor and the gliadin/glutenin ratio. Therefore, it is valuable to confirm the effect of gluten composition variation in O-free flour on cookie-making performance. 

When comparing O-free wheat to its two parent lines, Keumkang and Olgeuru, a deficiency of ω-5-gliadin was identified through RNA sequencing analysis, revealing a simple deletion in the chromosome. This finding was further corroborated by the similarity analysis using the expressed sequence tags (ESTs) for gliadins and glutenins. The wheat also exhibited a reduced low-molecular-weight glutenin subunit [4]. However, there has been few reports on the gluten protein composition of O-free flour in comparison to regular wheat flour using SDS-PAGE gel electrophoresis. It is valuable to confirm the gluten protein composition of O-free flour in relation to its performance in making cookies. 

Therefore, this study aims to analyze the quality characteristics of low-allergy O-free wheat flour by comparing its gluten protein composition, SRC, pasting, thermal properties, and dough-mixing properties with commercial flours of varying gluten strengths. It also evaluates the quality of cookies made with O-free flour.

## 2. Materials and Methods

### 2.1. Materials

The flour used in this study was milled from a low-allergy wheat cultivar, O-free, supplied by the National Institute of Crop Science (Wanju, Jeollabukdo, Republic of Korea). For a comparative study, three commercial flours with varied gluten strengths (strong, medium, and weak) were purchased from a local market: the flour with strong gluten strength (CJ Jeiljedang, Yangsan, Republic of Korea), the flour with medium gluten strength (referred to as all-purpose flour), and the flour with weak gluten strength (Samyang Corp., Asan, Republic of Korea).

### 2.2. Analysis of Moisture and Protein Content of Flour Samples

The moisture and ash contents of the wheat flour samples were measured according to Method 44-15.02 and 08-01,01, respectively [22]. For moisture content analysis, approximately 3 g of each flour was placed on pre-weighed aluminum containers and dried in a convection oven (FO 600-M, Jeio Tech Co., Ltd., Daejeon, Republic of Korea) at 130 °C for 1 h. The containers were then cooled in a desiccator for 30 min, and then the containers with the dried samples were reweighed. To assess ash content, approximately 3 g of each sample was weighed in a crucible. The crucibles were placed in a furnace at 550 °C (DMF-4.5.T, SciFinetech, Seoul, Republic of Korea) for 15 h. After cooling, the crucibles containing the samples were reweighed to determine the weight of the remaining inorganic materials.

The protein contents of the wheat flour samples were measured according to Method 46-30.01 [22]. The flour samples were combusted using an Elementar instrument (rapid N exceed, Elementar Analysensysteme GmbH, Langelselbold, Germany), and the nitrogen content was calculated. The crude protein content was determined in duplicate, as it exhibited good reproducibility, and it was calculated by multiplying the nitrogen content by the typical protein factor of wheat (5.7).

### 2.3. Extraction of Glutein Components and SDS-PAGE Gel Electrophoresis of Flours

Following the method outlined by Lookhart and Bean [23], albumin and globulins were extracted from 500 mg of the defatted sample using 5 mL of 0.15 M NaCl for 2 h. Subsequently, the mixture was centrifuged at 20,000× *g* for 15 min. Gliadin was then extracted from the remaining pellet by employing 5 mL of 70% ethanol for a duration of 12 h, followed by centrifugation under identical conditions. Glutenin, on the other hand, was extracted from the pellet after gliadin extraction, using a solution of 50% 1-propanol and 1% DTT for two hours, followed by centrifugation.

For the electrophoresis of the gliadin and glutenin extracts, a 4× Laemmli sample buffer containing 0.4 M DTT was combined with the protein extract in a 2:1 (*v*/*v*) ratio. A running buffer consisting of 10× Tris/Glycine/SDS Buffer was diluted tenfold and employed. Each extract (10 μL) was loaded onto 12% TGX Stain-Free Precast Gels, and electrophoresis was conducted at 70 V for 10 min, followed by 100 V for 90 min. Upon completion of electrophoresis, the staining solution (comprising 40% methanol, 10% acetic acid, and 0.1% Coomassie Brilliant Blue G-250) was applied for 30 min, followed by the destaining solution (40% methanol and 10% acetic acid) for 1 h. The extraction and electrophoresis experiments were conducted in triplicate.

### 2.4. Analysis of Solvent Retention Capacity of Flours in Four Solutions 

To evaluate flour quality, the SRC was determined according to Method 56-11.02 [22]. Four 50 mL conical tubes were weighed, and 5 g of each flour (5 g) was placed in each tube. Distilled water, 5% (*w*/*w*) lactic acid, 5% (*w*/*w*) sodium carbonate, and 50% (*w*/*w*) sucrose solutions were prepared, and 25 g of each solution was added to each tube containing the flours. The tubes were then shaken for 20 min at 5 min intervals to disperse and hydrate the flour and centrifuged at 1000× *g* for 15 min (LaboGene 1248, Gyrozen Inc., Daejeon, Republic of Korea). After centrifugation, the supernatant in each tube was discarded, and the pellet weight was measured to determine SRC percentages using the AACCI method [22].

### 2.5. Analysis of Thermal Characteristics of Flour Using Differential Scanning Calorimetry

Differential scanning calorimeter was used to analyze the gelatinization properties of starch in the flour with water, according to the method described by Kweon et al. [24]. Water or pre-dissolved sucrose solution (50% *w*/*w*) and flour were mixed in a 1:1 (*w*/*w*) ratio, and approximately 40 mg of the mixture was placed in an empty aluminum pan and sealed. A pan containing the sample was placed in a DSC instrument (DSC 6000, Penkin Elmer Co., Waltham, MA, USA) and heated at a rate of 10 °C/min from 10 to 140 °C. An empty aluminum pan was used as a reference, and the Pyris Software (ver. 11, Perkin Elmer Co., Ltd.) was used to calculate gelatinization temperatures and enthalpies.

### 2.6. Determination of Flour Pasting Property Using Rapid Visco-Analyzer

The pasting properties of starch in the flours were investigated using a RVA (RVA4, Newport Scientific, Sydney, Australia) based on the method described by Kweon et al. [24]. The RVA canister was filled with 25 mL of water or the pre-dissolved sucrose solution (50% *w*/*w*), and 3.5 g of the flour was added and thoroughly mixed using a plastic paddle. The RVA was run according to standard 1 profile (50–95 °C, 12.2 °C/min heating; 95 °C, 2.5 min maintained; 95–50 °C, 12.2 °C/min cooling; 50 °C, 2 min maintained). Pasting parameters such as peak, breakdown, final, and setback viscosities; peak time; and peak temperature were calculated using RVA software (Thermocline for Windows ver. 2.5).

### 2.7. Analysis of Dough-Mixing Property of Flour using Mixograph

A mixograph test was conducted using a 10 g Mixograph machine (National Manufacturing Inc., Lincoln, NE, USA). Each flour sample (10 g) was placed in a mixograph bowl, and 6.0–7.0 g of water or 6.5 g of pre-dissolved sucrose solution (50% *w*/*w*) was added. The Mixograph machine was run for 10 min, and the peak height and time of the graph were measured. Each sample was tested in duplicate.

### 2.8. Preparation of Cookies

Cookies were prepared following Method 10-53.01 [22], and the ingredients and formula are presented in Table 1. Shortening (90.0 g) and granulated sucrose (94.5 g) were weighed separately, placed in a bowl (N50; Hobart, Troy, OH, USA), and mixed for 3 min at speed 1 (low speed). Then, the corn syrup (3.4 g), water (49.5 g), and ammonia bicarbonate (1.1 g) were combined, placed in a bowl, and mixed at speed 1 for 1 min, followed by mixing at speed 2 (medium speed) for 1 min. Subsequently, skim milk powder (2.3 g), refined sea salt (2.8 g), and sodium bicarbonate (2.3 g) were thoroughly combined with flour (225.0 g), placed in a bowl, and mixed for 2 min at speed 1. After mixing, the dough was divided into four 60 g masses and placed on an aluminum baking pan. The dough was flattened using a rolling pin, and shaped using a circular cutter (6 cm in diameter). The dough was baked in an oven (Phantom M301 Combi; Samjung, Gyeonggi, Republic of Korea) at 215 °C for 12 min. After cooling the cookies for approximately 1 h at 25 °C, they were sealed in an aluminum foil bag, and their quality was assessed after 24 h.

### 2.9. Assessment of Cookie Quality

The length and width of each cookie were measured, and the average value was calculated using the measurements of the four cookies. Cookie height was measured using a caliper, arranging four cookies side by side vertically, and the average value was calculated from four measurements taken in different orders.

The top surface color of the cookies was measured by L* (lightness), a* (redness), and b* (yellowness) values using a chromometer (CR-20, Minolta, Co., Ltd., Tokyo, Japan), and the average value was obtained from four measurements. 

Cookie firmness was determined using a Texture Analyzer (CT3, Brookfield, Middleboro, MA, USA). was measured using a Texture Analyzer. The measurement conditions were as follows: Mode, measure force in compression; pre-test speed, 1.0 mm/s; test speed, 0.5 mm/s; probe, 2 mm cylinder probe; penetration distance, 10 mm. The measurement was repeated three times, and the average cookie firmness was calculated from these three measurements.

### 2.10. Statistical Analysis

All experiments, except for moisture, protein and ash content, were conducted in triplicate unless otherwise specified. All data were obtained through multiple measurements at least three times. The difference between the mean values of the samples was analyzed using analysis of variance with Tukey’s HSD test at a significance level of *p* < 0.05, using SPSS 22.0 (SPSS Inc., Armonk, New York, NY, USA) program.

## 3. Results and Discussion

### 3.1. Physicochemical Properties of Flour and SDS-PAGE gel Electrophoresis of Gluten Protein

The physicochemical properties of the flour samples are shown in Table 2. The moisture content of the flour samples was 13.08–14.0%, which is consistent with the typical range of flour moisture (12–14%) [25], indicating that the water added in the tempering process during the milling of the flour samples was not significantly different [24]. The ash content of the flour samples ranged from 0.37 to 0.43%; the ash content of the flour with weak gluten strength (W) was significantly lower than that of the other flour samples (*p* < 0.05), suggesting the lowest extraction rate during milling [26].

The protein content of the flour samples ranged from 8.50% to 13.98%. The flour with strong gluten strength (S) had the highest protein content, while the flour with weak gluten strength (W) had the lowest protein content. This is consistent with the relationship between flour protein and gluten strength, where flours milled from hard wheat (strong gluten) have higher protein content, and flours milled from soft wheat (weak gluten) have lower protein content [27]. The protein content of O-free flour (O) was 11.5%, which was lower than the flour with strong gluten strength but higher than the flours with medium and weak gluten strengths. This is in line with a previous study by Park et al. [28]. Kang et al. [16] and Ozturk et al. [29] reported that the flour protein content affected the cookie quality. When cookies are made with high-protein flour, he elasticity of the cookie dough increases, and the cookie spread decreases during baking, resulting in cookies with small diameters and large heights [21]. Therefore, flours with low protein content and weak gluten strength, such as the W flour sample, were suggested to be the more suitable for cookie production. 

The SDS-PAGE gel electrophoresis of glutenin and gliadin extracted from the flour samples was presented in Figure 1. In the glutenin and gliadin extract, SDS-PAGE gel electrophoresis of O-free flour exhibited significant differences compared to the other three flours. O-free flour displayed five distinct bands in the molecular weight range between 75 and 150 kDa, representing high-molecular-weight glutenin subunits (HWG-GS) [19,30,31], in contrast to the other flours which showed only three bands. Notably, glutenin extract exhibited more distinct bands in this molecular weight region compared to the gliadin extract. The intensity of these three bands in both extracts decreased in the order of S > M > W, reflecting the reduced amount of HWG-GS in the flour samples.

Importantly, no clear bands in the molecular weight range of 50–75 kDa were observed in the gluten extract, but approximately three bands appeared in the gliadin extract of S, M, and W flours, except for O-free flour. These results indicated the absence of ω-gliadins in O-free flour, which is consistent with findings from ω-gliadin-free wheat reported by Waga and Skoczowski [32]. The band intensity decreased in the order of S > M > W, indicating reduced amounts of ω-gliadins.

Additionally, the bands in O-free flour in the range of MW 37–50 kDa appeared significantly different from those of the other flours. Gliadin extract exhibited much thicker bands than glutenin extract, which could be attributed to differences in γ-gliadins and low-molecular-weight glutenin subunits (LMW-GS) [19]. In particular, the band near 50 kDa in both extracts was absent in O-free flour, indicating a reduction or absence of γ-gliadins and LMW-GS components. Meanwhile, the other flours exhibited a decrease in intensity similar to the bands of HWG-GS.

These results clearly demonstrate variations in the amount of glutenin and gliadin in the flours. Overall, the SDS-PAGE gel electrophoresis results confirm the differences in the gluten protein composition of O-free flour compared to flours with varying protein contents and gluten strengths [4]. Therefore, it can be assumed that the gluten protein composition and protein content of O-free flour may impact its performance in cookie making.

### 3.2. Solvent Retention Capacity of Flours

Figure 2 shows the solvent retention capacity (SRC) results of the flour samples in four different solutions: water, lactic acid, sodium carbonate, and sucrose. The SRC values indicate significant differences among the flour samples, following the order of flour with strong gluten strength > flour with medium gluten strength > O-free flour > flour with weak gluten strength (*p* < 0.05). This result was consistent with a study on the SRC of O-free flour conducted by Moon et al. [5].

Water SRC values for the flours with strong, medium, and weak gluten strengths were 72.9%, 64.8%, and 51.4%, respectively, while the O-free flour had a water SRC value of 55.4%. The flour with strong gluten strength had the highest water SRC value, indicating a potentially high amount of damaged starch generated during the milling of hard wheat. Water SRC values are associated with water absorption [33], suggesting that flours with weak gluten strength and O-free flour require less water to form machinable dough for making cookies. 

Lactic acid SRC is related to flour gluten strength [33]. The SRC values of the flours with strong, medium, and weak gluten strengths were 165.3%, 125.6%, and 91.4%, respectively, while the O-free flour had a lactic acid SRC value of 120.4%. These values indicate significant differences depending on the gluten strength. Notably, the O-free flour, which had higher protein content, exhibited a lower lactic acid SRC value compared to the flour with medium gluten strength. This can be attributed to the reduced low-molecular-weight glutenin protein content observed in O-free flour, as demonstrated in SDS-PAGE gel electrophoresis (Figure 1). This reduction in low-molecular-weight glutenin proteins has an impact on lactic acid SRC and overall gluten quality [4]. Among the flour samples, the flour with weak gluten strength was identified as the most suitable for cookie production, consistent with previous studies on cookie flour [29].

Sodium carbonate SRC is associated with damaged starch in wheat flour [33]. The SRC values of the flours with strong, medium, and weak gluten strengths were 102.2%, 89.2%, and 71.6%, respectively, while the O-free flour had a sodium carbonate SRC value of 85.6%. The flour with strong gluten strength showed a significantly high sodium carbonate SRC value due to the high kernel hardness of hard wheat, resulting in a high flour extraction yield and damaged starch during milling. Conversely, the flour with weak gluten strength exhibited a significantly low sodium carbonate SRC value due to the low kernel hardness of soft wheat, leading to a low extraction yield and damaged starch during milling [34].

Sucrose SRC is associated with the contribution of arabinoxylans in wheat flour [33]. The SRC values of the flours with strong, medium, and weak gluten strengths were 131.4%, 114.0%, and 95.1%, respectively, while the O-free flour had a sucrose SRC value of 104.9%. The results indicated a significantly higher arabinoxylan content in the flour with strong gluten strength compared to the flour with medium and weak gluten strengths and O-free flour. Arabinoxylans play a significant role in water absorption compared to damaged starch or gluten proteins [33]. Consequently, the poor cookie-making performance of the flour with strong gluten strength can be attributed to the challenge of moisture evaporation during baking, while the cookie-making performance of O-free flour may be superior to that of flours with strong and medium gluten strengths.

Gaines [35] reported that ideal cookies could be made with specific SRC values of wheat flour: water SRC ≤ 51%, lactic acid SRC ≥ 87%, sodium carbonate SRC ≤ 64%, and sucrose SRC ≤ 89%. Therefore, flours with low water, sodium carbonate, and sucrose SRC values allow for quick water evaporation during baking, making them favorable for cookie production [8,36]. In this context, the flour with weak gluten strength and the O-free flour, which exhibited relatively low SRC values in water, sodium carbonate, and sucrose solutions, are suitable for cookie production. Furthermore, the O-free flour is suitable for producing low-allergy cookies based on its solvent retention capacity. 

### 3.3. Gelatinization Properties of O-Free Flour in Water and Sucrose Solution Measured Using Differential Scanning Calorimetry (DSC)

The DSC results for the flour samples in both water and pre-dissolved 50% (*w*/*w*) sucrose solution are presented in Table 3. The pre-dissolved sucrose solution was employed to investigate the gelatinization of starch in the flour samples under conditions that mimic a sugar concentration found in cookie formulations. The gelatinization peak temperature (Tpeak) of the flours in water ranged from 68.2 to 70.8 °C, which was significantly lower than the Tpeak observed in the sucrose solution (ranging from 96.9 to 101.6 °C). This indicates a notable delay in the gelatinization of starch induced by sucrose solution (*p* < 0.05). This result is consistent with previous studies on the effect of sucrose solution on starch gelatinization [24,37,38,39].

Both the gelatinization peak and end temperatures of the flours, including those with strong, medium, and weak gluten strengths, as well as O-free flour, in water, exhibited values of 70.8, 69.2, 68.2, and 69.8 °C, respectively. Similarly, in the sucrose solution, these temperatures displayed a comparable trend across all flour samples. The gelatinization of starch in flour during the baking of cookies can significantly impact various attributes of the cookies, such as size, color, moisture content, and texture. Sucrose has been shown to delay the network formation of starch in flour during baking, leading to cookies with larger diameters and smaller heights [11]. These results suggest that the influence of flours with varying gluten strengths, as well as O-free flour, on cookie attributes related to starch gelatinization is likely to be similar.

### 3.4. Pasting Properties of O-Free Flour in Water and Sucrose Solution Measured Using Rapid Visco-Analyzer (RVA)

The RVA results for the flour samples in both water and pre-dissolved 50% (*w*/*w*) sucrose solutions are presented in Table 4. As previously utilized in the DSC analysis, the pre-dissolved sucrose solution, which simulated the cookie formulation, was employed to evaluate starch pasting in the flour samples within the RVA analysis. The pasting pattern of flours in water differed from that of flours in sucrose solution because the starch granules in water swelled quickly during heating to achieve the highest pasting viscosity. The swollen starch granules were then broken by continuous shearing, thereby reducing the viscosity. The viscosity increased again during cooling, indicating gel formation. In contrast, the pasting pattern of flours in sucrose solution showed a slow increase in viscosity because the starch granules did not swell quickly during heating. Furthermore, no noticeable peak was observed because the less swollen granules were not broken even after continuous shearing [24,38,40]. Notably, the pasting temperature of the flours in water (67.3–87.5 °C) was lower than that in the sucrose solution (90.7–92.8 °C). 

The higher peak viscosity was observed in water, ranging from 2454 to 3059 cP, com-pared to sucrose solution, ranging from 547 to 1742 cP. This indicates greater swelling of starch granules in water [41]. The breakdown viscosity, which reflects the breakdown of swollen starch granules and amylose release during continuous shearing, was higher in water (761–1221 cP) compared to sucrose solution (64–214 cP) [10,36]. The final viscosities of the flours in water (2752–3519 cP) were higher than those in sucrose solution (673–1987 cP). The setback viscosity, which indicates the binding of starch molecules and the formation of a gel network during cooling [42], was greater in water (1157–1644 cP) than in sucrose solution (189–543 cP), suggesting that starch aging and gel formation occurred more quickly in water. Further, the peak time to reach the highest viscosity was shorter in water (6.1–6.2 min) than in sucrose solution (6.9–7.0 min). 

Although there were variations in pasting viscosity values among the four flours when mixed with water, their pasting patterns displayed similarities. Additionally, similar trends were observed in the pasting properties of the flours when subjected to a sucrose solution. These results suggest that the pasting properties of O-free flour did not significantly differ from those of the other three flours, indicating that starch’s impact on cookie-making performance is minimal. The pasting viscosity values of O-free flour in a sucrose solution were relatively low, which presents a challenge when attempting to explain the results. Factors such as arabinoxylans, proteins, lipids, and particle size, in addition to starch in the wheat flour, may influence pasting viscosities.

### 3.5. Dough-Mixing Properties of Flour Samples

Mixograms of the flours in water and pre-dissolved 50% (*w*/*w*) sucrose solutions are depicted in Figure 3. The peak center height of the curve is a parameter indicating dough development and strength, which are associated with protein content and quality [43,44]. The flour with strong gluten strength exhibited a distinct peak at around 4 min, and the width of the dough-mixing band was maintained for approximately 10 min. The flour with medium gluten strength showed a broader peak, but the width of the dough-mixing band was still maintained. In contrast, the flour with weak gluten strength did not exhibit a peak. These mixograms are typical of flours with different gluten strengths [45]. The O-free flour displayed a distinct peak at approximately 2 min and 30 s, similar to that of the flour with strong gluten strength. However, the dough strength of the O-free flour weakened during continuous mixing and the width of the dough-mixing band narrowed, resembling the flour with weak gluten strength. Notably, the gluten network formed quickly and reached its maximum strength, but it collapsed and rapidly weakened when kneaded for a certain period.

Edwards et al. [46] and Khatkar et al. [47] analyzed the dough patterns of different flours with varying protein contents and quality. The wheat flour with strong gluten strength maintained a broad bandwidth even after reaching its maximum strength, while the wheat flour with weak gluten strength narrowed the bandwidth after reaching its maximum strength. These findings confirmed the effects of wheat flour quality and gluten protein content on dough properties. Furthermore, ω-gliadins were found to have the least positive effects on peak dough resistance in mixing characteristics and loaf volume [48]. 

Comparing the mixograms of flour in water to those of the flours in sucrose solutions, we observe that the latter did not display any distinct peaks. This absence of peaks indicates a delay in gluten development. Additionally, the flours with strong and medium gluten strengths exhibited broader bandwidths when contrasted with the flours with weak gluten strength and O-free flour. 

These results suggest that gluten development occurred more rapidly in the flours with strong and medium gluten strengths than in those with weak gluten strength. The accelerated development of gluten during cookie dough mixing resulted in reduced cookie spread and structural collapse, leading to smaller cookie diameters and greater cookie heights. Consequently, this variance in gluten development could potentially result in lower-quality cookies when using flours with stronger gluten characteristics. The quantity of glutenins and gliadins in the flour with weak gluten strength, along with the distinct gluten protein composition of O-free flour as illustrated in Figure 1, are crucial factors that likely influence their dough-mixing properties.

### 3.6. Cookie-Making Performance of Flour Samples

The top and side views of the cookies made using the various flour samples are depicted in Figure 4. The characteristics of the dough and cookies are shown in Table 5.

The width and length of the cookies followed the order: O-free flour ≥ flour with weak gluten strength > flour with medium gluten strength ≥ flour with strong gluten strength. This observation aligns with the predictions based on the solvent retention capacity (SRC) results of the flour samples (Table 2). Despite O-free flour having a higher protein content (Table 2), its gluten strength is relatively lower than that of flour with medium gluten strength, resulting in larger cookies. Typically, gluten imparts a viscoelastic property to dough by forming disulfide bonds between gliadin and glutenin [49,50]. O-free flour lacks ω-5 gliadin and has fewer low-molecular-weight glutenin subunits compared to the other three flours. In particular, ω-gliadin lacks sulfur and thiol groups, which are not involved in gluten development; however, low-molecular-weight glutenins contribute to gluten development [17,49]. Therefore, cookies prepared using O-free flour exhibited similar characteristics to those made with flour of weak gluten strength, indicating the suitability of O-free flour for producing high-quality cookies. The top surface colors of the cookies made with the different flour samples are also presented in Table 4. The cookies made with O-free flour did not significantly differ from those made with flour of weak gluten strength. A slight difference in b* of O-free cookies may be attributed to variations in protein composition and amino acids, leading to distinct color development through the Maillard reaction.

The cookie firmness followed the order: flour with weak gluten strength > O-free flour > flour with medium gluten strength > flour with strong gluten strength. Generally, cookies made from wheat flour with low protein content are known to exhibit high firmness and excellent crispness due to high moisture baked out [33]. This suggests that O-free flour can be used to create crispy cookies.

## 4. Conclusions

This study explores the potential of using O-free wheat, a low-allergy wheat cultivar, in cookie production to meet the demand for allergen-free and reduced-allergen foods. The research highlights the unique gluten composition of O-free flour, showcasing its absence of ω-gliadins and reduced levels of low-molecular-weight glutenins and γ-gliadins. The flour’s solvent retention capacity (SRC) places it between weak and medium gluten flours, emphasizing its distinctive quality. While thermal and pasting properties show similarities, mixograms reveal variations, indicating the critical role of gluten quality and composition in cookie dough characteristics. Cookies made with O-free flour closely resemble those produced with weak gluten flour, known for their appealing cookie attributes. In conclusion, this study supports using O-free flour for allergy-friendly cookies, providing a safe and tasty option for consumers with dietary needs.

## Figures and Tables

**Figure 1 foods-12-03843-f001:**
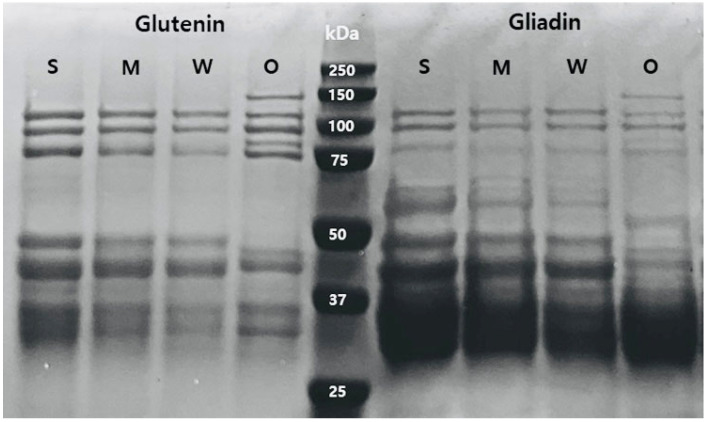
Sodium dodecyl sulfate-polyacrylamide gel electrophoresis of glutenin and gliadin extract from the flour with different gluten strengths and O-free flour: S, flour with strong gluten strength; M, flour with medium gluten strength; W, flour with weak gluten strength; O, O-free flour.

**Figure 2 foods-12-03843-f002:**
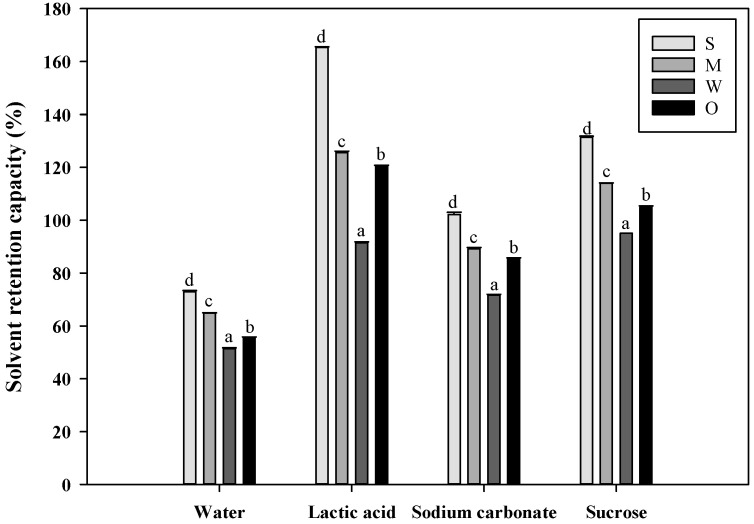
Solvent retention capacity (SRC) of the flour with different gluten strengths and O-free flour: S, flour with strong gluten strength; M, flour with medium gluten strength; W, flour with weak gluten strength; O, O-free flour. The same letters indicated above the bars are not significantly different at *p* = 0.05, according to Tukey’s HSD test.

**Figure 3 foods-12-03843-f003:**
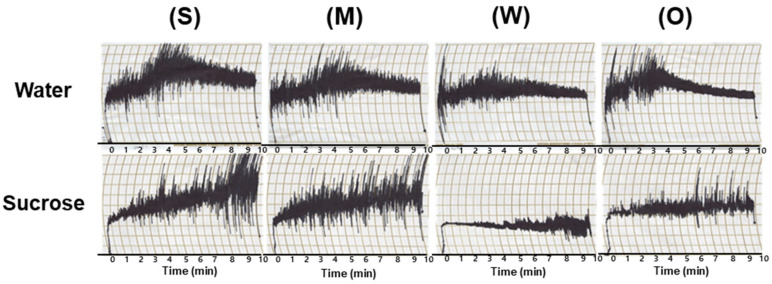
Mixograms of the flour with different gluten strengths and O-free flour in water and pre-dissolved 50% (*w*/*w*) sucrose solution. S, flour with strong gluten strength; M, flour with medium gluten strength; W, flours with weak gluten strength; O, O-free flour.

**Figure 4 foods-12-03843-f004:**
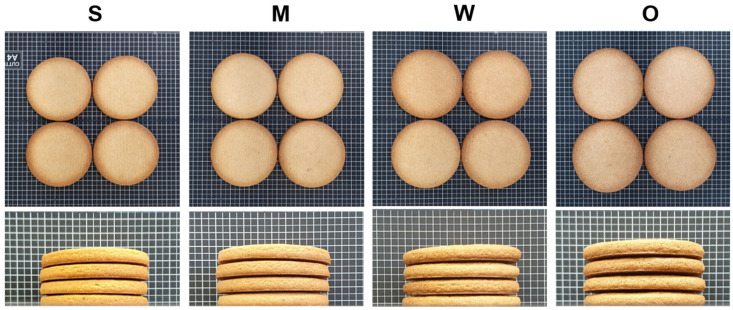
Top and side views of the cookies prepared using the flours with different gluten strengths and O-free flour: S, flour with strong gluten strength; M, flour with medium gluten strength; W, flour with weak gluten strength; O, O-free flour.

**Table 1 foods-12-03843-t001:** Ingredients and formula for AACC Method 10-53.01 cookie baking.

Ingredient	Standard Weight (g)
Flour	225.0 ^1^
Shortening	90.0
Sucrose	94.5
Corn syrup	3.4
Sodium bicarbonate	2.3
Non-fat dry milk	2.3
Salt	2.8
Ammonium bicarbonate	1.1
Water	49.5

^1^ Method 10-53 assumes 13% flour water content.

**Table 2 foods-12-03843-t002:** Moisture, ash, and protein contents, and SRC values of the flour with different gluten strengths and O-free flour.

Flour ^(1)^	Moisture (%)	Ash (%)	Protein ^(3)^ (%)
S	13.72 ± 0.03 ^c(2)^	0.42 ± 0.00 ^b^	13.98 ± 0.02 ^d^
M	13.50 ± 0.04 ^b^	0.43 ± 0.01 ^b^	10.56 ± 0.02 ^b^
W	14.01 ± 0.01 ^d^	0.37 ± 0.00 ^a^	8.50 ± 0.02 ^a^
O	13.10 ± 0.00 ^a^	0.43 ± 0.00 ^b^	11.49 ± 0.01 ^c^

^(1)^ S, M, and W, flours with strong, medium, and weak gluten strength, respectively; O, O-free flour. ^(2)^ Results are expressed as the mean ± SD. Values with the same letter within the same column are not significantly different (*p* < 0.05), according to Tukey’s HSD test. ^(3)^ Protein indicates the protein content adjusted for the moisture content of 14%.

**Table 3 foods-12-03843-t003:** Thermal characteristics of the flour samples in water and 50% (*w*/*w*) pre-dissolved sucrose solution.

Solvent ^(1)^	Flour ^(2)^	Tonset (°C)	Tpeak (°C)	Tend (°C)	Heat of Transition (ΔQ, J/g)
Water	S	60.8 ± 0.2 ^ab(3)^	70.8 ± 0.1 ^a^	90.6 ± 0.1 ^a^	3.0 ± 0.1 ^a^
	M	58.9 ± 0.2 ^a^	69.2 ± 0.5 ^a^	91.0 ± 0.6 ^a^	3.5 ± 0.1 ^ab^
	W	61.5 ± 0.1 ^b^	68.2 ± 0.0 ^a^	86.0 ± 0.3 ^a^	3.7 ± 0.0 ^bc^
	O	64.1 ± 0.3 ^c^	69.8 ± 0.3 ^a^	87.9 ± 1.3 ^a^	3.5 ± 0.1 ^ab^
Sucrose	S	88.5 ± 0.2 ^de^	101.6 ± 0.7 ^c^	120.4 ± 1.5 ^b^	3.4 ± 0.1 ^ab^
	M	86.3 ± 0.8 ^d^	98.3 ± 1.1 ^bc^	119.1 ± 1.5 ^b^	3.9 ± 0.1 ^bc^
	W	87.5 ± 0.2 ^d^	96.9 ± 0.3 ^b^	118.9 ± 0.2 ^b^	4.2 ± 0.0 ^c^
	O	89.8 ± 0.6 ^e^	98.2 ± 1.2 ^bc^	117.0 ± 3.2 ^b^	3.9 ± 0.2 ^bc^

^(1)^ Flour: solvent ratio = 1:1 (*w*/*w*). ^(2)^ S, M, and W, flours with strong, medium, and weak gluten strength, respectively; O, O-free flour. ^(3)^ Results are expressed as the mean ± SD. Values with the same letter within the same column are not significantly different (*p* < 0.05) according to Tukey’s HSD test.

**Table 4 foods-12-03843-t004:** Pasting characteristics of the flour samples in water and 50% (*w*/*w*) pre-dissolved sucrose solution.

Solvent ^(1)^	Flour ^(2)^	Peak Viscosity (cP)	Breakdown Viscosity (cP)	Final Viscosity (cP)	Setback Viscosity (cP)	Peak Time (min)	Pasting Temp (°C)
Water	S	3059 ± 18 ^g(3)^	1221 ± 19 ^g^	3482 ± 1 ^g^	1644 ± 3 ^f^	6.2 ± 0.0 ^a^	67.4 ± 0.2 ^a^
	M	2454 ± 3 ^e^	859 ± 1 ^e^	2752 ± 1 ^e^	1157 ± 1 ^d^	6.1 ± 0.1 ^a^	67.3 ± 0.4 ^a^
	W	2732 ± 21 ^f^	761 ± 8 ^d^	3519 ± 21 ^g^	1548 ± 8 ^e^	6.1 ± 0.0 ^a^	81.9 ± 0.4 ^b^
	O	2804 ± 13 ^f^	1022 ± 40 ^f^	3354 ± 21 ^f^	1572 ± 31 ^e^	6.2 ± 0.1 ^a^	87.5 ± 0.5 ^c^
Sucrose	S	1299 ± 23 ^c^	103 ± 4 ^ab^	1399 ± 23 ^b^	203 ± 4 ^a^	7.0 ± 0.0 ^b^	91.5 ± 0.5 ^d^
	M	1742 ± 7 ^d^	214 ± 4 ^c^	1987 ± 15 ^d^	459 ± 4 ^b^	7.0 ± 0.1 ^b^	90.7 ± 0.7 ^d^
	W	1177 ± 13 ^b^	187 ± 6 ^bc^	1533 ± 9 ^c^	543 ± 2 ^c^	7.0 ± 0.0 ^b^	92.0 ± 0.1 ^d^
	O	547 ± 0 ^a^	64 ± 2 ^a^	673 ± 4 ^a^	189 ± 6 ^a^	6.9 ± 0.1 ^b^	92.8 ± 0.0 ^d^

^(1)^ Flour: solvent ratio = 7:50 (*w*/*v*). ^(2)^ S, M, and W, flours with strong, medium, and weak gluten strength, respectively; O, O-free flour. ^(3)^ Results are displayed as the mean ± SD. According to Tukey’s HSD test, values with the same letter within the same column did not differ significantly (*p* < 0.05).

**Table 5 foods-12-03843-t005:** Geometry, top surface color and firmness of the cookies prepared using the flours with different gluten strengths and O-free flour.

Flour ^(1)^	Cookie Geometry (cm)			Top Surface Color			Cookie Firmness (N)
	Width	Length	Height	L*	a*	b*	
S	7.0 ± 0.1 ^a(2)^	7.0 ± 0.6 ^a^	0.94 ± 0.01 ^a^	65.1 ± 0.1 ^b^	12.5 ± 0.3 ^ab^	32.8 ± 0.2 ^b^	12.9 ± 0.1 ^a^
M	7.1 ± 0.8 ^a^	7.2 ± 0.3 ^a^	1.06 ± 0.01 ^b^	65.8 ± 0.2 ^b^	12.4 ± 0.3 ^a^	32.6 ± 0.3 ^b^	14.3 ± 0.2 ^b^
W	7.4 ± 0.6 ^b^	7.5 ± 0.4 ^b^	1.06 ± 0.01 ^b^	62.2 ± 0.4 ^a^	13.4 ± 0.2 ^b^	32.4 ± 0.5 ^b^	17.7 ± 2.0 ^d^
O	7.5 ± 0.6 ^b^	7.5 ± 1.0 ^b^	1.16 ± 0.01 ^c^	62.4 ± 0.1 ^a^	13.1 ± 0.1 ^ab^	31.0 ± 0.1 ^a^	16.5 ± 1.3 ^c^

^(1)^ S, M, and W, flours with strong, medium, and weak gluten strength, respectively; O, O-free flour. ^(2)^ Results are expressed as the mean ± SD. Values with the same letter within the same column are not significantly different (*p* < 0.05), according to Tukey’s HSD test.

## Data Availability

The data presented in this study are available upon request from the corresponding author.

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
