# Peer review of "Effect of Gluten Composition in Low-Allergy O-Free Wheat Flour on Cookie-Making Performance Compared with Flours with Different Gluten Strengths"

_foods, 2023, doi:10.3390/foods12203843_

Round 1

Reviewer 1 Report

Comments and Suggestions for Authors

Please explain what O-free flour is in the abstract when first mentioned.

Section 2.8  For true replicates, how many batches of each cookie formulation was prepared?

line 191  Was this three res from each cookie formulation batch or more reps than three each batch?

line 202 to 203  What do you mean by extraction rate?  If extraction rate means removing ash than W had the highest rate.

line 248 glutenin extract

line 295 to 296 yield of what?

line 315 to 316  Your O-flour values were greater than Gaines values for sucrose and sodium carbonate, will it still be good for cookies?

373 amylose release during

line 381 to 385  You need to explain why sucrose decreased pasting viscosities in O-flour much more than the other flours.

Fig 3  Can you add time on the x-axis so it is visible in the figure?

line 422 to 423  Be specific.  Is it faster development or slower development of gluten that results in lower quality?

line 435 to 436 Is 1/2 cm much larger even though its statistically different?  Would consumers notice?

line 444 to 446  O-flour cookies did differ in L* and b* values.  How do you explain that?

Comments on the Quality of English Language

Some minor grammar issues need correcting.

Author Response

Comments and Suggestions for Authors

Response) We appreciate very much the careful review and detailed comments given by the reviewer. We revised our manuscript marked with red color.

Please explain what O-free flour is in the abstract when first mentioned.

Response) O-free is a name of wheat cultivar, and we have revised the sentence to provide this explanation in the abstract.

Section 2.8  For true replicates, how many batches of each cookie formulation was prepared?

Response) We tested two or three batches of each cookie formulation.

line 191  Was this three res from each cookie formulation batch or more reps than three each batch?

Response) We conducted at least three measurements from each cookie formulation.

line 202 to 203  What do you mean by extraction rate?  If extraction rate means removing ash than W had the highest rate.

Response) Extraction rate means flour yield, which reflects the removal extent of bran. A higher extraction rate indicates less removal of bran, resulting in higher ash content. 

line 248 glutenin extract

Response) We corrected the typographical error.

line 295 to 296 yield of what?

Response) We have revised it to “flour extraction yield.”

line 315 to 316  Your O-flour values were greater than Gaines values for sucrose and sodium carbonate, will it still be good for cookies?

Response) Yes, O-free flour is suitable for cookies. Gaines values for sucrose and sodium carbonate is based on US standard cookie flour. The flour used in the study was milled in Korea, and W flour is generally used for producing cookies in Korea. Therefore, we compared O-free flour to the currently available cookie flour in Korea to assess its suitability for cookie production.

373 amylose release during

Response) We have corrected this as suggested.

line 381 to 385  You need to explain why sucrose decreased pasting viscosities in O-flour much more than the other flours.

Response) We acknowledge the difficulty in explaining this observation, despite our efforts.

Fig 3  Can you add time on the x-axis so it is visible in the figure?

Response) We have added time on the x-axis as requested.

line 422 to 423  Be specific.  Is it faster development or slower development of gluten that results in lower quality?

Response) We have added a specific explanation regarding gluten development.

line 435 to 436 Is 1/2 cm much larger even though its statistically different?  Would consumers notice?

Response) Although the difference in individual cookie size may appear small, it would be noticeable when aligning four cookies.

line 444 to 446  O-flour cookies did differ in L* and b* values.  How do you explain that?

Response) We have provided an explanation for the differences in only b* values, which was significantly different.

Comments on the Quality of English Language

Some minor grammar issues need correcting.

Response) We have made efforts to correct the grammar issues.

Reviewer 2 Report

Comments and Suggestions for Authors After reading the manuscript "Effect of gluten composition in low-allergy O-free wheat flour on cookie-making performance compared with the flours with  different gluten strengths", I realized that the manuscript showed in some parts the scientific rigour wanted, but in other parts I have missed it.

The authors have presented critical evaluation only in some paragraphs.

The references are not exactly current, besides the objective can be improved.

Thats why I have written some suggestions below in an attempt to improve the paper.

L.2- 3 "making performance compared with the flours with  different gluten strengths"-  I do not think it is necessary  some words in your title

L.31- This reference from 2018 seems to me not to be a good choice, since we have seen increasing numbers of coeliacs or gluten intolerants in the world. 

L.32- I'm sorry, but I didn't notice any relevance in mentioning peaches. If that were the case, you should have mentioned eggs, soya, peanuts... I'd mention cereals that contain gluten, and that is enough.

L.38- Check use of "the"

L.66- Check use of "the"

L.71-76 - Sorry, but objective is to long and repetitive. Please, improve it. "quality characteristics of low-allergy wheat O-free flour. Gluten protein composition through SDS-PAGE gel electrophoresis,  solvent retention capacity (SRC), pasting and thermal properties, and dough-mixing properties of O-free flour were compared with those of commercial flours with varying gluten strengths. Additionally, the quality of cookies prepared using O-free flour was  assessed" - 

Honestly, what I noticed after reading the whole paper were physical and chemical analyses. You could summarize it and make the objective clearer.

For example electroforesis is Methods, not here.

L.80- Where was O free flour purchased?

L.86- "Sucrose, shortening, skim milk powder, ammonium bicarbonate, baking soda, syrup, and  salt were also purchased from a local market. " - If in the next section you've only evaluated the flours and these ingredients are for the cookies, don't include them here, it will confuse the reader.

Furthermore, these ingredients need to be more detailed. Ex: what syrup ? sucrose ?  Refined salt? Himalayan Pink Salt? Flower of Salt? and so on ...

L.92- Which kind of oven ? Details, please.

L.99- Please, include whether they were evaluated in triplicate ? The same for all the sections.

L.101- I missed you quoting  authors for this methodology.

L.138- I missed you quoting authors for this methodology.

L.153- Is duplicate satisfatory  for this analysis?

L.155- The ingredients and quantities are always clearer and more understandable in a table, so if it's possible to include them, I strongly encourage it. Don't forget to include the details I suggested earlier.

L.165- Which oven ? 

L.166- "room temperature" is not a good option, please mention the numbers for real.

L.168- How many batches? How many replications?

L.171- "The hardness of the cookie" - check english, please.

L.178-179- It seems to me that you wanted to say that it had been measured. How did that happen? Caliper?

L.185- 188- In my opinion, this part should come close to L. 171-177- Also, check if these two paragraphs aren't repetitive.

L.225- Would it be possible to lighten this image a little and try to visualize it better? The bottom part is very dark.

L. 285- Did you evaluate the difference among the solvents?

I think you could make it clearer why you decided to evaluate only sucrose in Table 2 and Table 3. I ask this because in L.336 and 337 you mention that . "Sucrose leading to cookies with larger diameters and smaller heights" - I've been working with cookies for a few years and I don't consider these characteristics as positive.

L.418- 439 -"Gluten strength" is too repetitive in the reading, please improve it.

L.453- L.453- It got a bit confusing here.

Please, use the same nomenclature as for materials and methods. For example: Hardness? Where is this result ?

In the title of the table we have : Dough firmness, weight loss, and geometry of the cookies , and in the table are : Width, Length,  Height , L*

L.447- You haven't evaluated breaking strength ? why ?

L.460- After reorganizing the objective, you need to revise your conclusion

L.489 - Check Authors' Guide, especially for references-  some years are outside the rules.

Sorry, but your references are very old, you need to pay attention to that. I looked for 2019, 2020, 2021, 2022 and 2023 - only a few, please include more up-to-date references.

Comments on the Quality of English Language

Moderate editing of English language required

Author Response

Comments and Suggestions for Authors

After reading the manuscript "Effect of gluten composition in low-allergy O-free wheat flour on cookie-making performance compared with the flours with  different gluten strengths", I realized that the manuscript showed in some parts the scientific rigour wanted, but in other parts I have missed it.

The authors have presented critical evaluation only in some paragraphs.

The references are not exactly current, besides the objective can be improved.

Thats why I have written some suggestions below in an attempt to improve the paper.

Response) We appreciate very much the careful review and detailed comments given by the reviewer. We revised our manuscript marked with blue color.

L.2- 3 "making performance compared with the flours with  different gluten strengths"-  I do not think it is necessary  some words in your title

Response) We appreciate your opinion; however, we prefer to keep the title as it is because the comparison of gluten strength is a focal point.

L.31- This reference from 2018 seems to me not to be a good choice, since we have seen increasing numbers of coeliacs or gluten intolerants in the world.

Response) We have revised the sentence and changed the reference.

L.32- I'm sorry, but I didn't notice any relevance in mentioning peaches. If that were the case, you should have mentioned eggs, soya, peanuts... I'd mention cereals that contain gluten, and that is enough.

Response) We have eliminated the mention of peaches and revised the sentence accordingly.

L.38- Check use of "the"

Response) We have added "the" where necessary.

L.66- Check use of "the"

Response) We have added "the" where necessary.

L.71-76 - Sorry, but objective is to long and repetitive. Please, improve it. "quality characteristics of low-allergy wheat O-free flour. Gluten protein composition through SDS-PAGE gel electrophoresis,  solvent retention capacity (SRC), pasting and thermal properties, and dough-mixing properties of O-free flour were compared with those of commercial flours with varying gluten strengths. Additionally, the quality of cookies prepared using O-free flour was  assessed" -

Response) We have revised the objectives to make them more concise and clear.

Honestly, what I noticed after reading the whole paper were physical and chemical analyses. You could summarize it and make the objective clearer.

For example electroforesis is Methods, not here.

Response) We eliminated the electrophoresis.

L.80- Where was O free flour purchased?

Response) The "O-free" flour was supplied by a Korean government institute named the National Institute of Crop Science.

L.86- "Sucrose, shortening, skim milk powder, ammonium bicarbonate, baking soda, syrup, and  salt were also purchased from a local market. " - If in the next section you've only evaluated the flours and these ingredients are for the cookies, don't include them here, it will confuse the reader.

Response) We have eliminated the mention of ingredients as suggested.

Furthermore, these ingredients need to be more detailed. Ex: what syrup ? sucrose ?  Refined salt? Himalayan Pink Salt? Flower of Salt? and so on ...

Response) We detailed the ingredients.

L.92- Which kind of oven ? Details, please.

Response) The oven used was a convection oven, and we have added this information.

L.99- Please, include whether they were evaluated in triplicate ? The same for all the sections.

Response) We have included information about repeated measurements. Related to all the sections, we explained the repeated times in statistical analysis section.

L.101- I missed you quoting  authors for this methodology.

Response) We have cited the relevant authors for the methodology.

L.138- I missed you quoting authors for this methodology.

Response) We have cited the relevant authors for the methodology.

L.153- Is duplicate satisfatory  for this analysis?

Response) Yes, duplicate measurements are satisfactory when reproducibility is good.

L.155- The ingredients and quantities are always clearer and more understandable in a table, so if it's possible to include them, I strongly encourage it. Don't forget to include the details I suggested earlier.

Response) We have added a table (Table 1) to provide clearer information about the ingredients and quantities.

L.165- Which oven ?

Response) We have added the information about the type of oven used.

L.166- "room temperature" is not a good option, please mention the numbers for real.

Response) We have specified the temperature value.

L.168- How many batches? How many replications?

Response) We have provided the information that two batches were prepared, and three measurements were taken for each batch.

L.171- "The hardness of the cookie" - check english, please.

Response) We have changed "hardness" to "firmness."

L.178-179- It seems to me that you wanted to say that it had been measured. How did that happen? Caliper?

Response) We have clarified that the measurement involved arranging four cookies vertically and measuring the total height with a caliper.

L.185- 188- In my opinion, this part should come close to L. 171-177- Also, check if these two paragraphs aren't repetitive.

Response) We have removed one of the repetitive paragraphs.

L.225- Would it be possible to lighten this image a little and try to visualize it better? The bottom part is very dark.

Response) We have made adjustments to the image to improve its visibility.

  1. 285- Did you evaluate the difference among the solvents?

Response) We have measured SRC (solvent retention capacity) of each flour in four different solvents. SRC is a solvation test for flours based on the exaggerated swelling behavior of component polymer networks in selected individual diagnostic solvents (water, sodium carbonate, lactic acid, and sucrose associated with water absorption, damaged starch, gluten and arabinoxylans, respectively).

I think you could make it clearer why you decided to evaluate only sucrose in Table 2 and Table 3. I ask this because in L.336 and 337 you mention that . "Sucrose leading to cookies with larger diameters and smaller heights" - I've been working with cookies for a few years and I don't consider these characteristics as positive.

Response) We have provided an explanation for the choice to evaluate sucrose, emphasizing its impact on cookie dimensions and quality.

: Sucrose is a primary sugar used in cookie production, and it plays a crucial role in achieving good quality cookies. The ideal characteristics of cookies include a larger diameter and a smaller thickness (height). To attain these attributes, a high sucrose concentration in the cookie formula is essential. Sucrose acts as an anti-plasticizer, which means it suppresses gluten development during dough mixing and delays starch gelatinization during baking.

L.418- 439 -"Gluten strength" is too repetitive in the reading, please improve it.

Response) We have made efforts to revise and minimize the repetition of "gluten strength."

L.453- L.453- It got a bit confusing here.

Response) We have revised the text to enhance clarity.

Please, use the same nomenclature as for materials and methods. For example: Hardness? Where is this result ?

Response) We have made the necessary revisions for consistency and clarity.

In the title of the table we have : Dough firmness, weight loss, and geometry of the cookies , and in the table are : Width, Length,  Height , L*

Response) We have adjusted the title of the table for consistency.

L.447- You haven't evaluated breaking strength ? why ?

Response) We appreciate the question. In our laboratory, we did not measure breaking strength because geometric and color measurements were sufficient for assessing cookie quality.

L.460- After reorganizing the objective, you need to revise your conclusion

Response) We have revised the conclusion accordingly.

L.489 - Check Authors' Guide, especially for references-  some years are outside the rules.

Response) We have reviewed the Authors' Guide and made necessary adjustments for references.

Sorry, but your references are very old, you need to pay attention to that. I looked for 2019, 2020, 2021, 2022 and 2023 - only a few, please include more up-to-date references.

Response) The reason for using older references is that much of the foundational research on cookies has been conducted for a long time. Recent research in the field often focuses on the effects of adding functional or protein ingredients, which were not the primary focus of our study.

Comments on the Quality of English Language

Moderate editing of English language required

Response) The manuscript was edited by a professional editor in Editage (a company provides editing services).

Reviewer 3 Report

Comments and Suggestions for Authors

Abstract:

Line 11-12: „The increasing demand for allergen-free and reduced-allergen foods has led an investigation into the potential of O-free wheat flour with low allergenicity for cookie production” - - please define the abbreviation "O-free" when it appears after for the first time

Introduction:

-line 35-37: „To address this, the Rural Development Administration in Korea has recently developed a specialty wheat variety called O-free, which does not contain the main allergens…” – add details of what major allergens

Materials and Methods

- line 96-98: „The flour samples were combusted using an Elementar instrument…” – add the device type

Results:

-Table 1: to avoid a value of SD of ± 0.0, please provid the value of moisture, ash and protein content with the second decimal place; The same in the Table 3

lines 391-397 and Figure. 3: in the description of the results regarding the dough mixing properties (Fig. 3. Mixograms…), the Authors refer to time. Meanwhile, it is difficult for the reader to follow the course of the discussion because there is no time axis in Fig. 3 - I suggest adding it

Conclusions:

Line463-465: „The gluten protein composition, Solvent Retention Capacity (SRC), pasting and thermal characteristics, and dough mixing property of O-free ………….” – please remove this sentence because it is not needed in the conclusions part

Author Response

Comments and Suggestions for Authors

Response) We appreciate very much the careful review and detailed comments given by the reviewer. We revised our manuscript marked with green color.

Abstract:

Line 11-12: „The increasing demand for allergen-free and reduced-allergen foods has led an investigation into the potential of O-free wheat flour with low allergenicity for cookie production” - - please define the abbreviation "O-free" when it appears after for the first time

Response) "O-free" is the name of a wheat variety, and it doesn't have an abbreviation.

Introduction:

-line 35-37: „To address this, the Rural Development Administration in Korea has recently developed a specialty wheat variety called O-free, which does not contain the main allergens…” – add details of what major allergens

Response) We have added the major allergens for "O-free."

Materials and Methods

- line 96-98: „The flour samples were combusted using an Elementar instrument…” – add the device type

Response) We have added the product name of the Elementar instrument.

Results:

-Table 1: to avoid a value of SD of ± 0.0, please provid the value of moisture, ash and protein content with the second decimal place; The same in the Table 3

Response) We have provided the moisture and protein values with the second decimal place.

lines 391-397 and Figure. 3: in the description of the results regarding the dough mixing properties (Fig. 3. Mixograms…), the Authors refer to time. Meanwhile, it is difficult for the reader to follow the course of the discussion because there is no time axis in Fig. 3 - I suggest adding it

Response) We have added time on the x-axis in Figure 3.

Conclusions:

Line463-465: „The gluten protein composition, Solvent Retention Capacity (SRC), pasting and thermal characteristics, and dough mixing property of O-free ………….” – please remove this sentence because it is not needed in the conclusions part

Response) We have removed the sentence as suggested.

Round 2

Reviewer 2 Report

Comments and Suggestions for Authors

Dear authors, 

After another evaluation of the manuscript, I realized  improvement in the quality of the paper.  Even though, the authors have accepted only some  of my requests.

They  have improved English language, which is always useful to ask a native speaker for a final appreciation. They added more authors to better substantiate the methodology and corrected tables. Although they've added new authors, I still think the references could be more up-to-date. Readers have been demanding that we bring current ones.

This final paper is definitely much better, but I still think that the conclusion does not cover the paper. For example, the findings on the physical part I don't notice in the current conclusion.

Comments on the Quality of English Language

 Minor editing of English language required

Author Response

Dear authors,

After another evaluation of the manuscript, I realized improvement in the quality of the paper.  Even though, the authors have accepted only some of my requests.

They have improved English language, which is always useful to ask a native speaker for a final appreciation. They added more authors to better substantiate the methodology and corrected tables. Although they've added new authors, I still think the references could be more up-to-date. Readers have been demanding that we bring current ones.

Response) We appreciate very much the careful review and detailed comments again. We revised our manuscript marked with blue color.  We have updated the references the best of our ability.

This final paper is definitely much better, but I still think that the conclusion does not cover the paper. For example, the findings on the physical part I don't notice in the current conclusion.

Response) We have revised the conclusion as your comment.

Comments on the Quality of English Language

Minor editing of English language required

Response) We have made efforts to perform minor edits.